# Rare Hereditary Gynecological Cancer Syndromes

**DOI:** 10.3390/ijms23031563

**Published:** 2022-01-29

**Authors:** Takafumi Watanabe, Shu Soeda, Yuta Endo, Chikako Okabe, Tetsu Sato, Norihito Kamo, Makiko Ueda, Manabu Kojima, Shigenori Furukawa, Hidekazu Nishigori, Toshifumi Takahashi, Keiya Fujimori

**Affiliations:** 1Department of Obstetrics and Gynecology, Fukushima Medical University, Fukushima 960-1295, Japan; s-soeda@fmu.ac.jp (S.S.); yenyen@fmu.ac.jp (Y.E.); chika729@fmu.ac.jp (C.O.); tetsus@fmu.ac.jp (T.S.); k0810@fmu.ac.jp (N.K.); akabee60@fmu.ac.jp (M.U.); m2149@fmu.ac.jp (M.K.); s-furu@fmu.ac.jp (S.F.); fujimori@fmu.ac.jp (K.F.); 2Fukushima Medical Center for Children and Women, Fukushima Medical University, 1 Hikarigaoka, Fukushima 960-1295, Japan; nishigo@fmu.ac.jp (H.N.); totakaha@fmu.ac.jp (T.T.)

**Keywords:** rare hereditary gynecological cancer, Cowden syndrome, Peutz–Jeghers syndrome, DICER1 syndrome, rhabdoid tumor predisposition syndrome 2, molecular genetics

## Abstract

Hereditary cancer syndromes, which are characterized by onset at an early age and an increased risk of developing certain tumors, are caused by germline pathogenic variants in tumor suppressor genes and are mostly inherited in an autosomal dominant manner. Therefore, hereditary cancer syndromes have been used as powerful models to identify and characterize susceptibility genes associated with cancer. Furthermore, clarification of the association between genotypes and phenotypes in one disease has provided insights into the etiology of other seemingly different diseases. Molecular genetic discoveries from the study of hereditary cancer syndrome have not only changed the methods of diagnosis and management, but have also shed light on the molecular regulatory pathways that are important in the development and treatment of sporadic tumors. The main cancer susceptibility syndromes that involve gynecologic cancers include hereditary breast and ovarian cancer syndrome as well as Lynch syndrome. However, in addition to these two hereditary cancer syndromes, there are several other hereditary syndromes associated with gynecologic cancers. In the present review, we provide an overview of the clinical features, and discuss the molecular genetics, of four rare hereditary gynecological cancer syndromes; Cowden syndrome, Peutz-Jeghers syndrome, DICER1 syndrome and rhabdoid tumor predisposition syndrome 2.

## 1. Introduction

Approximately 5–10% of all malignancies are caused by hereditary cancer predisposition syndromes [1,2], and gynecological cancers originating in the uterus and ovary are also closely related to hereditary factors [3,4]. The most common and noteworthy hereditary gynecological cancer syndrome is hereditary breast and ovarian cancer syndrome (HBOC), wherein *BRCA1* and *BRCA2* germline pathogenic variants (PVs) have been identified. HBOC has been reported to occur in approximately one in 300 to one in 800 in the general population [5], and is a disorder marked by an increased lifetime risk of breast and ovarian cancers in women. More recent studies have improved our understanding of inherited ovarian cancer risk beyond *BRCA*-related HBOC, and have allowed more detailed estimates of ovarian cancer risk for several homologous recombination (HR)-related genes with moderate-penetrance such as *BRIP1*, *RAD51C/D*, *PALB2*, and *ATM* [6,7]. Poly(ADP-ribose) polymerase (PARP) inhibitors are strikingly toxic to cells with HR deficiency and comprise a promising therapeutic strategy for HBOC-related cancers [8]. Lynch syndrome, another notable hereditary gynecological cancer syndrome, occurs in approximately one in 400 individuals, and increases the risk of endometrial and ovarian cancers, as well as predisposing women to other, non-gynecologic cancers such as stomach, small intestine and colorectal cancers [2]. Germline PVs in one of four mismatch repair (MMR)-related genes called *MLH1*, *MSH2*, *MSH6* and *PMS2*, cause Lynch syndrome. Immune checkpoint inhibitor (ICI) has been proven to be a highly effective treatment strategy for microsatellite instability high (MSI-H) tumors associated with MMR-deficiency, such as Lynch syndrome [9]. Because of the high prevalence of both of these hereditary tumors, methods for screening, detection, management and treatment of additional malignancies associated with the syndromes have been established in detail.

In addition to HBOC and Lynch syndrome, there are also several gynecological cancers that occur in women with uncommon hereditary syndromes. Although the incidence of these syndromes is low, recent developments in genetic analysis using next-generation sequencers have identified several susceptibility genes in addition to the causative genes. Research and clinical trials are being conducted to target therapies for the cascade of related genes. The present review focuses on rare hereditary gynecological cancer syndromes, including Cowden syndrome (CS), Peutz-Jeghers syndrome (PJS), DICER1 syndrome and rhabdoid tumor predisposition syndrome (RTPS) 2, provides a summary of clinical features and relevant gynecologic cancers, and discusses molecular genetics. Table 1 shows an overview of molecular genetics for rare hereditary gynecological cancer syndromes.

## 2. Cowden Syndrome

### 2.1. Clinical Features

The reported frequency of CS is estimated to be approximately one in 200,000 [31]; however, the actual frequency may be higher than this estimate because clinical diagnosis is difficult. CS was first reported in 1963 by Lloyd and Dennis, who named the disease after their patient, Rachel Cowden [32]. CS is an autosomal dominant hereditary disorder characterized by the formation of hamartomas in various organs, including the skin, thyroid, breast, brain, and gastrointestinal tract; and the increased risk of the development of malignancy. CS has been reported to be associated with an increased lifetime risk of all forms of cancer (85–89%), particularly breast (77–85%), thyroid (21–38%), renal (15–34%), endometrial (19–28%), and colorectal (9–16%) cancers [33,34,35]. The National Comprehensive Cancer Network (NCCN) has modified the original consortium clinical criteria to include family history and major/minor criteria to provide diagnostic support [36]. Additionally, both the NCCN and Cleveland clinic have established genetic testing guidelines for PTEN hamartoma tumor syndrome (PHTS)/CS. CS is part of the PHTS, a group of disorders caused by PVs in the PTEN gene. Other clinical syndromes that are part of the PTEN hamartoma tumor syndrome are Bannayan-Riley-Ruvalcaba (BRR) syndrome, Proteus syndrome (PS), and Proteus-like syndrome. Germline *PTEN* PVs occur in up to 85% of CS/CS-like individuals who meet International Cowden Consortium operational diagnostic criteria, in up to 60% of BRRS cases, 7–20% of PS cases and 50–67% of Proteus-like syndrome cases [37].

### 2.2. Gynecologic Cancers

Although endometrial cancer (EC) has been recognized as a key component of CS, its prevalence and clinical features are not well characterized. Strong clinical predictors for the presence of germline PTEN variants in EC in CS/CS-like patients suggest an age of under 50 years, macrocephaly, high phenotypic burden, and/or coexistence with renal cell carcinoma [38]. Patients with PHTS have 40 times the risk of EC compared to the general population, a lifetime cumulative risk of 19–28% by age 70 years, and a median age of 48 years at EC diagnosis, which is 20 years younger than the general population of 68 [39]. Screening for EC is more controversial, as it has not been shown to reduce mortality; however, additional imaging, endometrial biopsy, or risk-reducing hysterectomy after completion of child-bearing may be considered on an individual basis. If surveillance for EC is offered, screening through endometrial biopsy every 1–3 years should start at 35 years old [36].

### 2.3. Molecular Genetics

*PTEN* is located at chromosome 10q23.3, encoding for a 403-amino acid protein that possesses both lipid and protein phosphatase activities. Since the PI3K/Akt/mTOR signaling pathway results in the regulation of signal transduction and biological processes such as cell proliferation, apoptosis, metabolism and angiogenesis, an increased in this pathway activity is a major contributor to human cancer. The PI3K/Akt/mTOR signaling pathway is tightly regulated by PTEN, which acts as a negative regulator of the pathways by converting phosphatidylinositol 3,4,5-trisphosphate back to phosphatidylinositol 4,5-bisphosphate. Inactivation of PTEN by loss-of-function mutations leads to elevated levels of phosphatidylinositol 3,4,5-trisphosphate and accelerates tumor initiation and progression. Various genetic alterations, including point mutations, large chromosomal deletions, and epigenetic mechanisms frequently cause PTEN inactivation in a large array of cancers [40]. Although the germline variants spectrum in PHTS is broad, with mutations affecting all nine exons of *PTEN*, approximately two-thirds of germline *PTEN* PVs occur in exons 5, 7, and 8, encoding the phosphatase domain [41]. In particular, exon 5 is a hotspot for germline variants because of its catalytic core motif, accounting for about 40% of *PTEN* variants [42,43]. Approximately 10% of CS patients have *PTEN* promoter variants in the germline, in addition to intragenic mutations [44]. Recently, it has been reported that some intronic variants of PTEN cause pathogenic exon skipping, alternative splicing, or the use of cryptic splice sites [45]. Large deletions of PTEN throughout the coding sequence were detected in 3–10% of PHTS patients [41,44,46]. Of note, the existence of gain-of-function variants of germline *AKT1* (2.2%) and *PIK3CA* (8.8%) associated with the PI3K/AKT/mTOR pathway was reported in PTEN wild-type CS/CS-like patients [47].

Since PTEN negatively regulates the PI3K/AKT/mTOR pathway, inhibition of mTOR is a reasonable therapeutic strategy, but its use is still limited to clinical trials. A phase II open-label clinical trial involving 18 CS patients showed that administration of oral mTOR-inhibitor (sirolimus) was tolerable in terms of its side effect profile, and was sufficient to down-regulate mTOR signaling in skin and gastrointestinal tissue [10]. A double-blind drug-placebo, crossover study with another mTOR inhibitor (everolimus) is currently being conducted in PHTS patients with autism spectrum disorder in the USA (NCT02461446). In addition to mTOR inhibition, upstream components of the PTEN signaling pathway, such as PI3K and AKT, also serve as candidates for pharmacologic inhibition in *PTEN*-mutated disorders. Recently, promising studies have been reported, in which a pan-AKT inhibitor (ARQ 092) was used to treat primary fibroblasts in vitro in patients with PIK3CA-associated overgrowth spectrum (PROS), and the PIK3CA inhibitor BYL719 (Alperisib) was used to treat a preclinical mouse model of PROS, followed by 19 patients with severe PROS disorders. [11,12]. BYL719 reduced the size of intractable vascular tumors and improved congestive heart failure, hemihypertrophy and scoliosis, with a promising safety profile, in patients with PROS regardless of PIK3CA mutation type [12].

Beyond *PTEN*, several germline PVs have been reported to exhibit the CS phenotype. The *KLLN* gene localizes to 10q23 and shares a transcription start site with *PTEN*. *KLLN*, a target gene of the tumor suppressor p53, is involved in cell cycle arrest and apoptosis. Germline hypermethylation in the promoter accounts for up to 35% of *PTEN* mutation-negative CS/CS-like patients, and is associated with increased prevalence of breast and renal cell carcinomas [48,49].

The succinate dehydrogenase complex (SDHx) is a multipart enzyme made of four subunits encoded by *SDHA*, *SDHB*, *SDHC* and *SDHD* genes, and is assembled in the mitochondria to form the mitochondrial complex 2, a key respiratory enzyme which links the Krebs cycle and the electron transport chain. Mutations in the *SDHx* genes are associated with a predisposition for developing hereditary pheochromocytoma and/or paraganglioma, GIST and Renal cell carcinoma [50]. Approximately 10% of *PTEN* wild-type CS/CS-like patients had germline *SDHB/D* PVs, which were associated with an increased prevalence of thyroid cancer compared with germline *PTEN* PVs alone [51].

The *SEC23B* gene provides instructions for making one component of a large group of interacting proteins called coat protein complex II and regulates the transportation of proteins and lipids from the endoplasmic reticulum to the Golgi apparatus in cells [52]. Previous studies have reported that alteration of SEC23B is associated with the development of thyroid cancer, colorectal cancer, and prostate cancer [53,54,55]. In a study of 96 CS and CS-like affected patients with thyroid cancer patients, germline heterozygous *SEC23B* variants were observed in three patients (3%) [56].

WWP1 (WW domain containing E3 ubiquitin protein ligase 1) is a member of the NEDD4-like family of HECT ubiquitin ligases, and plays important roles in a diverse variety of biochemical and cellular processes, such as protein degradation, transcriptional regulation, cell proliferation and differentiation, apoptosis, and senescence. Overexpression of WWP1 has been identified in a variety of cancers, and is associated with poor prognosis [57]. A recent study reported that WWP1 activation in animal and in vitro models inhibited PTEN function, which led to protumorigenic phenotypes [58]. A cohort study indicated that germline heterozygous *WWP1* variants were observed in five of 126 patients (4%) from a study population consisting of CS and CS-like affected patients with wild-type *PTEN* who had gastrointestinal oligopolyposis as a predominant phenotype [59].

A study evaluating 371 EC patients with CS or CS-like clinical picture found germline *PTEN* mutations in 7% of patients, germline *SDHx* variants in 9.8%, and *KLLN* promoter hypermethylation in 10.5% of patients [60]. Although studies to elucidate the etiology of non-PTEN-related CS have identified *SDHx*, *KLLN*, *AKT1*, *PIK3CA*, *SEC23B*, and *WWP1* as candidate genes for predisposition to CS, these genes have not yet been validated for use in the clinical setting. Identifying and accumulating data on susceptibility genes are important for patients to receive gene-specific genetic counseling, predictive testing of family members, and accurate risk assessment.

## 3. Peutz-Jeghers Syndrome

### 3.1. Clinical Features

PJS is a relatively rare disorder with an estimated incidence of about one in 25,000 to one in 280,000 births and is an autosomal dominant disorder characterized by melanocytic macules of the lips, buccal mucosa, and digits; multiple gastrointestinal hamartomatous polyps [61]. The clinical diagnosis of PJS has been established in a proband with one of the following: (1) two or more histologically confirmed PJ polyps; (2) any number of PJS-type polyps detected in an individual who has a family history of PJS in at least one close relative; (3) characteristic mucocutaneous pigmentation in an individual who has a family history of PJS in at least one close relative; and (4) any number of PJS-type polyps in an individual who also has characteristic mucocutaneous pigmentation [61,62]. Individuals with PJS are at increased risk of multiple malignancies. The cumulative cancer risk in such individuals has been estimated to be 76–93%, with a median age of 45 years at first cancer diagnosis [63,64]. In addition, the risks of cancer by organ are 45–50% for breast carcinoma, 39% for colorectal carcinoma, 29% for gastric carcinoma, 13% for small intestinal carcinoma, 11–36% for pancreatic carcinoma, and 15–17% for lung carcinoma [65]. Due to the increased risk of various malignancies, diligent surveillance is recommended [66].

### 3.2. Gynecologic Cancers

Ovarian sex cord tumor with annular tubules (SCTATs) is one of the gynecological cancers associated with PJS. Although SCTATs are very rare neoplasms comprising less than 1% of sex cord ovarian tumors, it has been reported that 36% (27/74) of SCTAT is complicated by PJS [67]. Compared to sporadic SCTAT, familial SCTATs generally occur in younger patients, are bilateral, small in diameter, multifocal and calcified, and have a favorable prognosis [67]. Although transvaginal ultrasound can be considered to begin at age 18–20 years, screening for SCTATs in women with PJS is controversial, given the lack of evidence in reducing mortality due to ovarian cancer [68].

Another PJS-related gynecological cancer is minimal deviation adenocarcinoma (MDA), which occurs in the cervix. Although MDA is a very rare malignancy, accounting for less than 1% of uterine cervical cancer, the incidence of MDA in PJS patients is estimated to be 15–30%, while up to 10% of MDA cases are complicated by PJS [69]. The mean age of reported MDA patients who developed PJS was 33 years, which was younger than the mean age of reported patients without PJS, which was 55 years [70]. As for surveillance of MDA, annual cervical cancer screening including pap smears starting at age 21 is recommended for women with PJS [71].

### 3.3. Molecular Genetics

PJS is a rare autosomal dominant inherited disorder caused by a germline PV in the *STK11* gene, also known as the *LKB1*, which is located on 19p13.3 and comprises nine coding exons and one noncoding exon, coding a member of the serine/threonine kinase family with 433 amino acids. *STK11* is known as a tumor suppressor gene and is an essential serine/threonine kinase that regulates various cellular processes, including cell metabolism, cell polarity, and apoptosis. The catalytic kinase domain is highly conserved in STK11 protein, which is comprised of amino acid residues in 49–309 positions. Although 80–85% of individuals with PJS have a mutation detected by sequencing, the remaining 15–20% have large rearrangements, deletions and duplications [61]. Hence, long-range PCR, multiplex ligation-dependent probe amplification (MLPA) and a gene-targeted microarray are also needed for genetic diagnosis. Most germline STK11 PVs in PJS patients are frameshift or nonsense changes, which result in an abnormal truncated protein and the consequent loss of kinase activity. Loss-of-function of STK11 leads to PJS phenotypes. Germline *STK11* PVs have been identified in up to 80% of familial PJS cases; for the remaining patients, PJS likely resulted from de novo mutations [72]. A small number of cases of mosaicism have been reported [73,74]. Loss of heterozygosity is typically observed in emerging cancers in patients with germline alterations.

In the past few decades, somatic mutations of *STK11* have been identified in many sporadic malignancies [75]. The predominant types of alterations are mutation and deletion of the *STK11* gene (as in lung adenocarcinoma and cervical squamous cell carcinoma), followed by amplification (as in sarcomas) [76]. STK11/LKB1 phosphorylates AMPK, leading to phosphorylation of TSC1/2, which then reduces Rheb activity, thereby limiting activation of mTOR [77], an effector protein complex for cell growth and survival pathways. Loss of STK11 therefore reduces AMPK and TSC1/2 phosphorylation, releasing inhibition of Rheb and allowing activation of mTOR, thereby promoting tumorigenesis.

Several approaches have been proposed to target STK11-deficient tumors based on preclinical observations. A case study reported that a PJS patient with *STK11*-mutated pancreatic cancer experienced partial responses to mTOR inhibitor (everolimus) [78]. However, retrospective analysis of a phase II trial for patients with endometrial carcinoma found that LKB1 protein levels were not significantly correlated with response to everolimus treatment [79]. Tumors with *STK11* mutations are also associated with T cell excluded tumors, which are characterized by low or no PD-L1, low T-cell densities, high levels of granulocyte colony stimulating factor and IL-8 family cytokines, high density of neutrophil-like cells, and production of myeloid cell-recruiting chemokines such as IL-6 [80,81,82]. Some studies have reported that *STK11* alteration is associated with poor response to immune checkpoint inhibitors (ICIs) in patients with non-small cell lung cancer, including those with tumors harboring co-occurring *KRAS* mutation [82,83,84].

Although there have been few effective treatments targeting *STK11* to date, several ongoing clinical trials are being conducted. STK11 is involved in DNA damage response, promoting homologous recombination and fostering genomic stability by interacting with *BRCA1* [13]. Two clinical trials using PARP inhibitors such as olaparib (NCT03375307) and talazoparib with avelumab (NCT04173507) are ongoing. Since STK11 and AMPK have been shown to be involved in the maintenance of redox homeostasis and reactive oxygen species (ROS)-induced cancer cell death, the functioning of the STK11-AMPK pathway may be a negative predictor of response to ROS-induced therapies [85]. STK11 has also been shown to function as a negative regulator of cellular ROS stress. Therefore, STK11-deficient tumors are characterized by uncontrolled cellular proliferation with elevated energetic stress, which increases intracellular ROS concentrations. Galan-Cobo et al. demonstrated that glutaminase inhibition blocks cell proliferation and increases energetic and oxidative stress sensitivity in STK11-deficient tumors [14]. A clinical trial is underway to offer the glutaminase inhibitor CB-839 hydrochloride for treatment of metastatic or unresectable solid tumors harboring mutations in several genes involved in metabolism, oxidative stress, and mTOR regulation that includes STK11 (NCT03872427).

## 4. DICER1 Syndrome

### 4.1. Clinical Features

DICER1 syndrome is an autosomal dominant manner caused by a germline *DICER1* PV, an important component of the microRNA processing pathway [86]. The hallmark neoplasm of DICER1 syndrome is pleuropulmonary blastoma (PPB), a pediatric dysembryonic sarcoma of the lung and pleura [87,88,89]. Although PPB is a rare tumor, accounting for 0.25–0.5% of malignant lung neoplasms, and typically presents in infants and children younger than 6 years old, germline PVs of *DICER1* have been reported to be found in about 70% of PPB cases [90,91]. Aside from PPB, several other characteristic neoplasms have been reported in *DICER1* mutation carriers, including cystic nephroma, rhabdomyosarcoma, multinodular goiter and thyroid carcinoma, which often develops in childhood [87,88,89]. Excluding tumors, macrocephaly is the most common clinical finding. In a study on macrocephaly associated with the DICER1 syndrome, the number of subjects with a head occipitofrontal circumference above the 97th percentile was significantly higher in patients with *DICER1* PVs (28/67, 42%) than in family controls (5/43, 12%) [92].

### 4.2. Gynecologic Cancers

Sertoli-Leydig cell tumor (SLCT), a rare ovarian tumor that belongs to the group of sex cord stromal tumors, accounts for less than 0.5% of ovarian tumors, and is the second most frequent DICER1-related tumor after PPB. Approximately 60% of SLCTs occur against a background of PVs in the germline of *DICER1* [93]. Compared with SLCT patients without DICER1 mutations, those harboring DICER1 mutations have features associated with androgen action, early onset, and frequent recurrence [94,95]. De Paolis et al. reported a median age at diagnosis of 17.6 years in 46 patients with SLCT with germline DICER PVs [96], which is younger than the previously reported age of onset (approximately 30 years) [97,98]. Prognosis of ovarian SLCTs is closely related to the clinical stage and degree of tumor differentiation [99]. A study focusing on patients with moderately or poorly differentiated SLCTs reported that patients with germline *DICER1* variants were more likely to exhibit clinical relapse, although this tendency was not statistically significant [100].

Embryonal rhabdomyosarcoma of the uterine cervix incidence is a very rare form of cancer that develops in childhood, and is reported to account for about 0.4–1% of all cervical malignancies [101]. *DICER1* somatic alterations are identified in 65–95% of uterine embryonal rhabdomyosarcomas, with up to 50% of these patients harboring germline alterations [102,103]. Gynandroblastoma is a rare subtype of SCST with a combination of female (granulosa cells) and male (Sertoli and/or Leydig cells) sex cord differentiation. *DICER1* hot-spot mutation is the key-driving event in a subset of gynandroblastomas containing components of SLCT and juvenile granulosa cell tumor [104,105]. The mean age of gynandroblastoma patients with *DICER1* germline PVs has been reported to be 16 years old [86].

### 4.3. Molecular Genetics

The DICER1 gene, located on chromosome 14q32.13, has 27 exons and encodes the endoribonuclease Dicer protein of the ribonuclease III family. DICER1, as RNase III, is known to be involved in the regulation of post-transcriptional gene expression through the production of microRNAs, as well as in the generation of siRNAs that regulate gene expression. In individuals with PPB with a detectable germline *DICER1* PVs, approximately 80% of the germline PVs were inherited from a parent, with the remaining 20% were de novo [106,107]. Germline PVs reported in *DICER1* include deletions, duplications, insertions, transitions, or transversions, as well as point mutations [89]. The majority of identified germline variants are located within regions that encode one of Dicer’s seven defined domains (helicase domains, the Dicer dimerization domain [DDD], the Piwi/Argonaute/Zwille [PAZ] domain, the RNase III domains, and the double-stranded RNA-binding domain) [89]. In a previous study, mosaic *DICER1* variants were also reported to be associated with *DICER1* syndrome in patients with severe phenotypes [108]. Germline sequencing identified a mosaic germline *DICER1* PV in 2% of the allelic reads in the patient’s blood sample [86]. Despite the rarity of most DICER1 syndrome tumors, germline *DICER1* PV is estimated to occur in the general population in approximately one in 10,600 [109]. A recent study, using the exome sequence data of 92,296 participants, estimated germline DICER1 putative loss-of-function variant prevalence to range from one in 3700 to one in 4600 people [110].

Tumors that develop as DICER1 syndrome typically have a germline nonsense or frameshift PV in one allele, and a somatic missense mutation in the other allele. Since RNase IIIa cleaves the 3p hairpin arm and RNase IIIb cleaves the 5p hairpin arm, the DICER1 cancer hotspot mutant has a selective defect in processing the miRNA-5p strand, resulting in an overall decrease in the 5p/3p ratios. Somatic mutations found in tumors of DICER1 syndrome affect five “hot spot” codons, encoding critical amino acids in the metal-binding catalytic cleft of the RNase IIIb domain: E1705, D1709, G1809, D1810, and E1813 [90]. These mutations inhibit the canonical processing of miRNA precursors, resulting in a relative excess of 3p-miRNAs and depletion of 5p-miRNAs [111]. This results in abnormalities in gene expression, organogenesis, and cell proliferation that are regulated by miRNAs, leading to tumorigenesis [112,113]. Recently, evolutionary and structural coupling analyses revealed that the RNase IIIa-S1344 site is located in close proximity to the active cleft of the RNase IIIb domain, and mutations in RNase IIIa-S1344 indicate the same pattern of 5p-miRNA loss as that caused by RNase IIIb hotspot mutations [114].

Some studies have demonstrated that metformin can upregulate DICER1 in mouse and human cells [15,16,17]. Although the treatments may not be beneficial for patients with biallelic *DICER1* mutations, those with a single functional DICER1 allele may benefit from metformin or similar compounds [89]. A DICER1-related syndrome with the acronym GLOW (global developmental-delay lung cysts-overgrowth-Wilms tumor) is caused by mosaic missense hotspot mutations in DICER1 affecting the RNase IIIb domain [115]. Recently, it has been reported that hotspot mutations in the RNAse IIIb domain are associated with dysregulation of specific miRNAs, leading to activation of the PI3K/AKT/mTOR pathway, and GLOW syndrome in concurrence with DICER1 syndrome may benefit from the use of PI3K/AKT/mTOR inhibitors such as rapamycin and TORIN-1 [18].

## 5. Rhabdoid Tumor Predisposition Syndrome 2

### 5.1. Clinical Features

RTPS are rare, aggressive malignancies, typically diagnosed in infants and children aged younger than 3 years. The origins of Rhabdoid tumors (RTs) are the central nervous system (65%), kidneys (9%), and, in the remaining 26% of cases, the soft tissues of the head and neck, paravertebral muscles, liver, bladder, mediastinum, retroperitoneum and pelvis [116]. RTPS involving the central nervous system is referred to as an atypical teratoid/rhabdoid tumor (AT/RT), and up to 50% of cases occur in the cerebellum [117]. RTPS is an autosomal dominant syndrome associated with an increased risk of RTs [118,119]. Patients with a germline variant in the *SMARCB1* gene have RTPS1, whereas those with a germline variant in *SMARCA4* have RTPS2. The majority of RTs show alterations that result in loss of SMARCB1 function, and a small proportion of RTs carry inactivated SMARCA4 [120]. The percentage of RTPS caused by germline PVs has been reported to be 85–95% for *SMARCB1* and 5–15% for *SMARCA4*, respectively [121].

### 5.2. Gynecologic Cancers

Patients with RTPS associated with gynecologic cancers are typically *SMARCA4* deficient (RTPS2). One of the most common cancers in RTPS2 is small-cell carcinoma of the ovary, hypercalcemic type (SCCOHT), which is a rare and aggressive cancer that mainly occurs in adolescent and young women. This tumor has a poor prognosis, with overall survival of less than two years in most cases [122]. Sequencing of additional tumors, familial and sporadic in origin, revealed that all had loss of SMARCA4 expression resulting either from mutations or loss of heterozygosity, identifying SCCOHT as a monogenic disease. Since somatic SMARCA4 mutations have been reported to be present in over 90% of SCCOHT cases using next-generation sequencing, they likely serve as the driver mutation for almost all cases [123,124]. To date, there have been a total of 118 unique somatic SMARCA4 mutations found in SCCOHT cases, consisting of frameshift mutations (43/118, 36.4%), stop/nonsense (38/118. 32.2%), splice-site (24/118, 20.3%), missense (7/118, 5.9%), and inframe deletion (6/118, 5.1%) alterations [125]. In a study of 60 patients, 26 carrying germline *SMARCA4* PVs were diagnosed at a significantly younger median age (21.5 years) than the 34 non-carriers (25.5 years, *p* = 0.02) [126]. Familial cases were often bilateral [127,128]. It is likely that up to 40% of women with SCCOHT may have germline variants in *SMARCA4* [129], so the detection of young and bilateral SCCOHT can be suggestive of RTPS2 [128]. Other gynecological cancers, such as undifferentiated uterine sarcoma, have also been described as RTPS2-related tumors [130].

### 5.3. Molecular Genetics

The *SMARCA4* gene is located in the chromosome 19p13.2, and encodes 1647 amino acids, a 185kDa catalytic subunit of switch/sucrose-non-fermenting (SWI/SNF). The SWI/SNF complex is an evolutionarily conserved ATP-dependent chromatin remodeling complex that plays important roles in DNA repair, transcriptional activation of genes normally repressed by chromatin, differentiation, and organ development, and has been shown to behave as a tumor suppressing complex [131]. Each SWI/SNF complex is comprised of multiple subunits that have an ATP-dependent catalytic unit, either SMARCA4 (BRG1) or SMARCA2 (BRM), as well as a core regulatory subunit such as SMARCB1 (BAF47/INI1), and various other subunits such as ARID1A [132,133].

Germline variants in subunits of the BRG1/BRM-associated factor (BAF) or SWI/SNF chromatin reassembly complex have been revealed in benign and malignant tumors and neurodevelopmental disorders [134]. A rare genetic disorder associated with the *SMARCA4* and *SMARCB* genes is known as Coffin-Siris syndrome (CSS). It is a congenital multi-systemic genetic disorder characterized by intellectual disability, coarse facial features, hypoplasia or absence of the fifth fingernail or toenail, and multiple other abnormalities, including ocular features [135]. Mutations in genes that encode proteins of the SWI/SNF complex, called the BAF complex in mammals, have been found to be responsible for CSS by whole exome sequencing and pathway-based genetic testing [136]. Some studies have reported that germline variants in *SMARCB1* and *SMARCA4* were involved in the CSS phenotype [137,138]. Germline PVs of *SMARCB1* (*n* = 14, 7%) and *SMARCA4* (*n* = 32, 15%) were also reported to be identified among 208 CSS patients [139].

Recent sequencing studies have identified somatic mutations in subunits of the SWI/SNF chromatin remodeling complexes in over 20% of human cancers [140]. In particular, SMARCA4 is one of the most frequently aberrant chromatin remodeling ATPases in cancer, being altered in approximately 5–7% of all human malignancies; moreover, *SMARCA4* genomic alterations and loss of SMARCA4 expression have been observed in some tumors [141]. Some reviews have reported several possible target therapies, including clinical trials for SMARA4-deficient tumors such as SCCOHT [125,141,142].

Although little is known about the efficacy of ICIs for SMARCA4-alterated cancers, some case reports for patients with relapsed SCCOHT, SMARCA4-deficient thoracic sarcoma and SMACA4-altered non-small cell lung cancer have been published [19,20,21]. Furthermore, ICI treatment correlated with significantly improved overall outcomes in SMARCA4-aberrant non-small cell lung cancer [22]. In these reports, there was no correlation between tumor mutation burden and/or IHC as a biomarker of efficacy with ICI and response to ICI. Although SCCOHT has a low tumor mutation burden [123], the immunogenic microenvironment resembling the landscape of the tumors with immune checkpoint blockade may provide the rationale for immunotherapy. A phase II basket trial with pembrolizumab (NCT03012620) is currently open for women with rare ovarian tumors, including relapsed SCCOHT.

Dysfunction of the SWI/SNF complex, due to loss of SMARCA4, leads to oncogenic dependence on EZH2 through transcriptional repression caused by aberrant trimethylated lysine 27 of histone H3. A subset of SMARCA4-deficient tumors was found to be sensitive to EZH2 inhibition, which serves as the catalytic subunit of polycomb repressive complex 2 [23]. The only study in SCCOHT patients (NCT02601950) is examining the most studied EZH2 inhibitor, tazemetostat (EPZ-6438), and early results from these phase I/II studies reported two patients with SCCOHT (one with stable disease and one with partial response after treatment).

Lysine-specific histone demethylase 1 (LSD1) regulates the chromatin landscape and gene expression by demethylating proteins such as histone H3, and is highly expressed in SWI/SNF-mutated tumors in the TCGA database. Soldi et al. showed that SWI/SNF-deficient ovarian cancer appears to be dependent on LSD1 activity, because SCCOHT and ovarian clear cell carcinoma cell lines are sensitive to SP-2577 (Seclidemstat), a reversible LSD1 inhibitor [24]. A phase I trial (NCT04611139) of Seclidemstat with PD-1 antibody (pembrolizumab) is ongoing for patients with recurrent SCCOHT as well as select additional ovarian and endometrial cancers with mutations in the genes within the SWI/SNF pathway.

Xue et al. found that inactivation of SMARCA4 causes significant downregulation of cyclin D1, limits CDK4/6 kinase activity in SCCOHT cells, and results in sensitivity to CDK4/6 inhibitors in vitro and in vivo [25]. They also observed SMARCA4-deficient non-small cell lung cancer results in reduced cyclin D1 expression, as well as selective sensitivity to CDK4/6 inhibitors [26]. Based on the above findings, the Canadian Cancer Trial Group has added a new arm to the ongoing CAPTUR trial (NCT03297606) to treat SMARCA4-deficient cancers using CDK4/6 inhibitor (palbociclib). A phase I clinical trial using another CDK4/6 inhibitor (abemaciclib) is currently being conducted in children and young adults with newly diagnosed diffuse intrinsic pontine glioma and recurrent/refractory solid tumors including RTs (NCT02644460).

BET inhibitors bind to the bromodomains of the bromodomain and extra-terminal motif (BET) proteins BRD2, BRD3, BRD4, and BRDT, and act to prevent protein–protein interaction between BET proteins and acetylated histones and transcription factors. SCCOHT cells were highly sensitive to BET inhibitors such as JQ1 and OTX-015, and the latter in particular showed strong antitumor activity in an orthotopic xenograft model of SCCOHT [27]. A recent study reported that cell migration through the downregulation of SMARCA4 was suppressed by BET inhibitor treatment in hepatocellular carcinoma cell lines [28].

Histone deacetylases (HDACs) are expressed at increased levels in cells of various malignancies, and HDACs inhibitors have shown clinical benefits in hematological malignancies but failed in solid tumors due to the lack of biomarker-driven stratification [143]. Although HDAC inhibitors restore SMARCA2 expression, which strongly suppresses growth of SCCOHT cells and showed in vivo sensitivity of SCCOHT cells to the HDAC inhibitor (quisinostat) [29], a report of a single case did not find efficacy with this approach [144].

SWI/SNF components, including SMARCA4, have also been implicated in DNA-damage repair [145]. SMARCA4-deficient lung adenocarcinoma cells showed increased DNA replication stress and greater sensitivity to the ATR inhibitor in vitro and in vivo [146].

Aurora kinase A (AURKA) is a serine/threonine kinase involved in a number of central biological processes, such as the G2/M transition, mitotic spindle assembly, and DNA replication. Since AURKA activity has been identified as essential for survival and proliferation in non-small cell lung cancer cells lacking SMARCA4, AURKA inhibitors may provide a therapeutic strategy for biomarker-driven clinical studies to treat the non-small cell lung cancers harboring SMARCA4-inactivating mutations [30].

## 6. Conclusions

Four rare hereditary gynecological cancer syndromes have been described in the review. The challenges faced by people at high risk of certain cancers include the development and validation of better methods for cancer detection and prevention. An understanding of the genetic causes and molecular pathways of hereditary cancer syndromes can inform our knowledge and potential new treatment of various types of cancers.

## Figures and Tables

**Table 1 ijms-23-01563-t001:** Overview of molecular genetics for rare hereditary gynecological cancer syndromes.

Hereditary Cancer Syndromes	Related Cancers ^1^ (%)	Affected Genes		Potential Targets	Drugs	Available Clinical Trials	References
2Cowden	Breast (77–85)	*PTEN* (~85%)					
	Thyroid (21–38)	*KLLN*	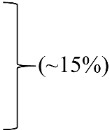	mTOR	Everolimus, Sirolimus	NCT02461446	[10]
	Kidney (15–34)	*SDHx*		AKT	ARQ092		[11]
	Colon (9–16)	*SEC23B*		PIK3CA	BYL719		[12]
	**Endometrium (19–28)**	*WWP1*					
Peutz-Jeghers	Breast (45–50)	*STK11*					
	Colon (36)						
	Gastric (29)						
	Small intestine (13)			PARP	Olaparib, Talazoparib	NCT03375307, NCT04173507	[13]
	Pancreas (11–36)			Glutaminase	CB-839 HCl	NCT03872427	[14]
	Lung (15–17)						
	**SCTAT (36)**						
	**MDA (15–30)**						
DICER1	PPB (70)	*DICER1*		Metformin	Metformin		[15,16,17]
	**SLCT (60)**			PI3K/ATK/mTOR	Rapamycin, TORIN-1		[18]
	**ERMS (~50)**						
RTPS2		*SMARC4A*		PD-1	Pembrolizumab	NCT03012620	[19,20,21,22]
				EZH2	Tazemetostat	NCT02601950	[23]
	CNS (65)			LSD1	Seclidemstat	NCT04611139	[24]
	Kidney (~7)			CDK4/6	Abemaciclib, Palbociclib	NCT02644460, NCT03297606	[25,26]
	**SCCOHT (~40)**			BET	JQ1, OTX-015		[27,28]
				HDAC	Quisinostat		[29]
				AURKA	VX-680		[30]

^1^ Related gynecological cancers were shown in bold style. Abbreviations: RTPS, rhabdoid tumor predisposition syndrome; SCTAT, ovarian sex cord tumor with annular tubule; MDA, minimal deviation adenocarcinoma; SLCT, Sertoli-Leydig cell tumor; ERMS, embryonal rhabdomyosarcoma; CNS, central nervous system; SCCOHT, small-cell carcinoma of the ovary, hypercalcemic type; PARP, poly(ADP-ribose) polymerase.

## Data Availability

Not applicable.

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
