# Peer review of "Rare Hereditary Gynecological Cancer Syndromes"

_ijms, 2022, doi:10.3390/ijms23031563_

Round 1

Reviewer 1 Report

This review article gives an overview of hereditary syndromes that contribute to gynecological tumors, including Cowden syndrome, Peutz-Jeghers syndrome, DICER1 syndrome, and RTPS. The manuscript is well written with a nice structure and organization that makes it easy to read. However, there are a few areas that could be improved to make the manuscript more informative.

  1. Lines 44-46: Do the authors mean that these mutations cause Lynch syndrome instead of are caused by Lynch syndrome?
  2. Section 2.3: could the authors provide more specifics about how these PTEN mutations affect its function or expression? How such defects would be expected to affect cell function?
  3. Lines 123-127: could the authors briefly describe any of the findings from these studies?
  4. Section 3.3: Compared to the description of the PTEN mutations, details on STK11 gene mutations seem to be lacking. Can the authors please provide more details about whether or not there are any recurrent PVs, how these PVs affect gene function, etc?
  5. Line 239: is there any evidence that PJS tumors have high levels of numerical or structural aneuploidy?
  6. Section 4.3: the first paragraph of this section is difficult to read and there are multiple typos/awkward sentences that should be revised. 
  7. Line 355: is should be replaced with are
  8. Line 468-469: awkward sentence, please revise

Author Response

Response to Reviewer 1 Comments

Thank you for your review of our paper. We have replied to each comment, point by point, below.

  1. Lines 44-46: Do the authors mean that these mutations cause Lynch syndrome instead of are caused by Lynch syndrome?

Response

Thank you for pointing this out. According to your comment, we have now changed “are caused by” to “cause” (P2, lines 46).

  1. Section 2.3: could the authors provide more specifics about how these PTEN mutations affect its function or expression? How such defects would be expected to affect cell function?

Response

Thank you for pointing this out. We have made changes and added some explanatory sentences regarding this point to Section 2.3 (P3, lines 100–110).

  1. Lines 123-127: could the authors briefly describe any of the findings from these studies?

Response

Thank you for pointing this out. Although research on the clinical effects of ARQ 092 in PROs patients is still ongoing, the clinical effects of BYL719 in PROS patients are described in reference 29. We have added a sentence regarding the details of the clincal effects and safety of BYL719 treatment (P3, lines 135-137).

  1. Section 3.3: Compared to the description of the PTEN mutations, details on STK11 gene mutations seem to be lacking. Can the authors please provide more details about whether or not there are any recurrent PVs, how these PVs affect gene function, etc?

Response

Thank you for pointing this out. We have added three sentences regarding this point to  Section 3.3 (P5, lines 219–226).

  1. Line 239: is there any evidence that PJS tumors have high levels of numerical or structural aneuploidy?

Response

Thank you for raising this important issue. Although PJS tumors have not been reported to have high levels of numerical or structural aneuploidy, several papers have reported that STK11 contributes to genome stability (ref. 67, PMID: 23584481 and PMID: 32668413).

  1. Section 4.3: the first paragraph of this section is difficult to read and there are multiple typos/awkward sentences that should be revised. 

Response

Thank you for pointing this out. We have made changes and added some explanatory sentences regarding this point to the first paragraph of Section 4.3.

  1. Line 355: is should be replaced with are

Response

Thank you for pointing this out. We can not find “is” in line 355. Is line 355 correct?

  1. Line 468-469: awkward sentence, please revise

Response

Thank you for pointing this out. We had simplified the description and it has been revised to be more detailed (P10, line 488-492).

Reviewer 2 Report

Comment 1.

In the manuscript, the authors explained the genes underlying Rare Hereditary Gynecological Cancer Syndromes. As you know, each of those pathogenic germline variants have a penetrance value, which predict the possibility of causing such cancer types. The authors should add a subsection into the manuscript, where they should explain by what the penetrance of pathogenic variants is determined.   

Author Response

Response to Reviewer 2 Comment

Thank you for your review of our paper. We have responded to your comment, below.

Comment 1: In the manuscript, the authors explained the genes underlying Rare Hereditary Gynecological Cancer Syndromes. As you know, each of those pathogenic germline variants have a penetrance value, which predict the possibility of causing such cancer types. The authors should add a subsection into the manuscript, where they should explain by what the penetrance of pathogenic variants is determined.  

Response

Thank you for raising this important issue. We agree that explaining how the penetrance of pathogenic variants is determined is important. However, since other genetic and environmental factors are expected to play roles in the presentation of clinical phenotypes and disease onset in hereditary cancer syndromes, it is very difficult to explain how the penetrance of pathogenic mutations is determined, especially in rare hereditary tumors. The interaction between environmental factors and the rare hereditary gynecologic cancer syndromes presented in this paper has not been quantified to date.